# The Influence of the Provenance and Spatial Structure on the Growth of European Silver Fir (*Abies alba* Mill.) of Autochthonous Origin in a Forest Plantation in the Białowieża Forest

Aleh Marozau [1,*], Uladzimir Kotszan [2], Justyna Nowakowska [3,*], Daria Berezovska [4], Warren Keith Moser [5], Tom Hsiang [6] and Tomasz Oszako [7]

1 Institute of Forest Sciences, Bialystok University of Technology, Wiejska 45E Street, 15-351 Bialystok, Poland
2 Faculty of Forestry, Belarusian State Technological University, Sverdlova Street 13a, 220006 Minsk, Belarus; wolodia250@mail.ru
3 Institute of Biological Sciences, Cardinal Stefan Wyszynski University in Warsaw, Woycickiego 1/3 Street, 01-938 Warsaw, Poland
4 Department of Biochemistry and Pharmacogenomics, Faculty of Pharmacy, Medical University of Warsaw, 02-097 Warsaw, Poland; dberezovska@wum.edu.pl
5 Rocky Mountain Research Station, Forest Service, U.S. Department of Agriculture, 2500 S. Pine Knoll Dr., Flagstaff, AZ 86001, USA; warren.k.moser@usda.gov
6 Environmental Sciences, University of Guelph, Guelph, ON N1G 2W1, Canada; thsiang@uoguelph.ca
7 Forest Protection Department, Forest Research Institute, Braci Lesnej 3, 05-090 Sekocin Stary, Poland; t.oszako@ibles.waw.pl
* Correspondence: a.marozau@pb.edu.pl (A.M.); j.nowakowska@uksw.edu.pl (J.N.)

**Abstract:** Currently, a change in the species composition of the Białowieża Forest, eastern Poland, is occurring. Parallel to the dying of spruce (*Picea abies* L. Karst.), caused by *Ips typographus* (L.), there is a spread of deciduous tree species, among which hornbeam (*Carpinus betulus* L.) is the most active. Against the background of climate change, it is of interest to consider the possibility of reintroducing relict species in areas occupied until recently by spruce. One of these is silver fir (*Abies alba* Mill.), which is biologically and ecologically very similar to spruce. The Tisovik tract (the Belarusian part of the Białowieża Forest) is the most northeastern and the only preserved refuge of autochthonous silver fir in the region. Therefore, it is the most suitable source of propagation of this species outside the mountainous part of its range. The target area of our study was a 26-year old artificial stand included in the information system of forest genetic resources conserved in Europe. It was created in 1996 in the Polish part of the Białowieża Forest in the Hajnówka Forest District (Forestry Wilczy Jar) from 10 half-sib families originating from seeds collected in the Tisovik tract. The goal of the study was to assess the influence of provenance factors and spatial structure on productivity and phenotypic variation as measured by diameter (DBH). The study's initial results showed that the provenance factor's influence at the pole wood stage manifests itself more clearly than spatial structure. Even within the framework of individual half-sibs, there was a clear differentiation of diameters between investigated trees. Interpretation of the obtained results allowed us to conclude that there is a "threshold value" of the distance between trees in a stand of a given age, which determines the point when its increase no longer contributes to an increase in DBH. The most promising phenotypes were selected for further research and practical actions to reproduce silver fir further.

**Keywords:** Białowieża Forest; autochthonous silver fir; half-sib plantation; placement of trees; DBH

## 1. Introduction

Since 2012, there has been extensive mortality of the Norway spruce (*Picea abies* L. Karst.) due to insect attacks [1,2]. The most harmful pest is the European spruce or eight-toothed bark beetle (*Ips typographus* L.)—a secondary pest that primarily affects weakened

and depressed trees. A change in the hydrological regime due to global warming and acid rain facilitated the weakening of spruce. It should be borne in mind that the spruce in the Białowieża Forest in Poland (which means further Forest) is located in the south of the northeastern part of the range [1]. Therefore, it is especially susceptible to changes in hydrology. The bark beetle actively attacks even healthy and robust trees during mass reproduction. This negative phenomenon occurs in the Forest, in Poland, including areas with rich soil conditions that fully meet the ecological and biological requirements of *A. alba* [3–5]. This species is natural to the Forest but has survived only in a small refugium, in the Tisovik tract in the Belarusian part of the Forest [6,7]. It lies 500 km northeast of the edge of the continuous range of silver fir in Poland and 120 km from the nearest natural island habitat in the Jata Reserve [8]. At present, it is the rarest coniferous species in the Forest.

Taking into account the autochthonous nature of the silver fir from Tisovik, it is reasonable to assume that the *A. alba* does not pose a threat to biodiversity, but, on the contrary, enhances it. This is unlike, for example, a very aggressive invasive species, *Quercus rubra* (L.), which displaces *Q. robur* (L.) and thus poses a real threat to the stability of forest ecosystems [9,10].

It is interesting to study the possibility of including silver fir as a candidate for artificial reforestation in the Forest and adjacent forest districts. According to our previous research, silver fir has exhibited high growth rates and vigor in stands of different ages and origins [11,12]. These performance measures justify establishing a seed base of silver fir for targeted afforestation of those areas freed up by spruce mortality.

A half-sib family is a collection of plants obtained from seeds from free pollination from one tree. In this context, the study of the unique gene pool of autochthonous silver fir, presented in the form of half-sib families (what we understand as "provenance") in the Hajnówka Forest District (FD), is necessary to identify the most promising genotypes. These promising genotypes could be preferred seed sources, especially for establishing seed plantations. Therefore, the studies of the first stage [11,12], continued in this research, were conducted using traditional forest mensuration and phenotypic analyses.

It is essential to note that despite its small size—approximately 20 trees at present [7,13,14]—the autochthonous population of silver fir in the Tisovik tract is similar in the degree of polymorphism and heterozygosity (based on isoenzymatic and DNA analyses) compared to the populations from the main distribution area in the Carpathians [15]. The list of publications on the genetics of fir in the Carpathians, without comparison with the population from Tisovik, is quite extensive [16–18]. Unfortunately, only this publication provides a genetic comparison of the population from Tisovik with populations from the main area. This small population shows no inbreeding effect [15], suggesting the value of a detailed genetic study of half-sib families, especially in selecting candidates for preferred seed trees. A study by Pawlaczyk et al. [19] showed an excess of heterozygotes on a half-sib plantation in the Hajnówka FD, which is the offspring of such a small population as the fir group in the Tisovik tract. The authors attribute this to a "bottleneck effect" and assume that pollination of the remnant firs in Tisovik could have been carried out, among other things, from two stands (artificial, of unknown geographical origin) in the Polish part of the Białowieża Forest. Based on genetic analysis, Mejnartowicz [20] became convinced that these two stands are not descendants of autochthonous fir from the Tisovik tract (even though the nearest stand is located 8.5 km in a straight line from this refuge). If the assumption of Pawlaczyk et al. [19] is correct, then not only the progeny of local cross-pollinated silver fir from Tisovik is represented in the half-sib plantation, but also individuals that arose as a result of gene flow together with pollen from these two stands of unknown origin, which are located in two forest subcompartments, 453Aa and 498 Ci of the Białowieża FD. We were unsuccessful in finding documents on the history of their appearance (between 1920 and 1939) in Polish archives, likely due to the destruction occurring during the Second World War. They might exist in some archives in Belarus, which are not available to us at the moment.

It would be logical to assume that there is a hereditary high degree of differentiation of DBH (diameter at breast height) between the families of the studied silver fir half-sibs and within families. Confirming or refuting this assumption was one of our research objectives, presented below. Note that at this stage, the genetic factor is conventionally represented as the membership of trees in one of the families of half-sibs (provenance). In addition, only the next step will be a study based on the evaluation of DNA polymorphism of the identified best specimens for this phenotype (DBH) trait.

Competitive interactions between trees occur at all stages of the forest stand dynamics, but they are the most pronounced at the second—stem—exclusion stage of forest formation [21]. The density and spacing of the trees determine the competitive influence experienced by any single tree. These interactions, in turn, influence the spatial structure of the stand, affecting both productivity and species diversity of forest communities. Optimising the spatial structure of forest stands as a function of age classes, and taking into account the biological characteristics of the breed is one of the most effective methods of increasing a stand's productivity and stability. Thus, regulating the spatial distribution of trees is the most important tool for managing artificial forest ecosystems [22].

The development of stand growth modeling raises the question of how to simulate the initial states of these models and, in particular, how to simulate realistic spatial structures [23]. The parameters of the spatial structure of forest stands determine and characterize them quantitatively. However, only understanding how to change the spatial structure effectively is crucial for forest management [24]. Knowledge of the spatial structure of stands can be used to develop silvicultural regimes to increase the area of semi-natural forests established as even-aged plantations [25]. Growth models of stands considering their spatial characteristics can predict productivity and evaluate changes in forest fund parameters and are recommended for use in forestry [26].

Thus, the aim of this study was a comparative evaluation of the influence of the provenance factor and spatial structure on the growth of trees in a 22-year-old artificial stand of autochthonous silver fir from the Tisovik tract.

The objectives of this study were as follows:

I. For the provenance factor:
    a.  to evaluate the variability of the most important silvicultural parameter—DBH—and thus reveal the degree of possible differentiation both between the families of the studied silver fir half-sibs and within families;
    b.  to identify the most productive phenotypes, which are potential sources of vegetative and seed material for effective reforestation of the autochthonous population of silver fir and objects of further genetic studies;

II. For the spatial structure factor:
    a.  to determine the parameters of the spatial structure effectively affecting the DBH,
    b.  to determine the approaches of effective regulation of the spatial structure of the stand in the near term.

## 2. Materials and Methods

### 2.1. Object of Research

The bulk of the study was conducted in 2018 in a half-sib plantation represented by ten families of silver fir from the Tisovik tract, located in the Belarusian part of the Białowieża Forest. The plantation was established by Professor A. Korczyk in Hajnówka FD (subcompartment 416Ag (now 416Ad)) in rich soil conditions (mixed fresh forest) in 1996 with 4-year-old seedlings grown from seeds [12]. The half-sib number corresponds to the number of the tree in the refugium. Unfortunately, according to our information, some mother trees have died for various reasons. However, they are presented in the form of half-sib families at the research object. The arrangement of plants at planting in furrows created in a recent cutting is 1.3 × 1.0 m. A total of 1551 seedlings (7692 specimens/ha) were planted in 65 rows of 27 m length (north-south), numbered in ascending order from

west to east. The half-sibs located in the first, oldest plot were examined. At the time of the study, it had been 22 years since the trees were planted, and their biological age was 26 years. The average diameter of the trees in the stand at the time of the study was 10.0 cm, and the average height was 9.4 m [12]. The other two plots, planted in 1998, are included in our future research plans under a scientific grant. All three plots are fenced to protect them from animals (Figure 1).

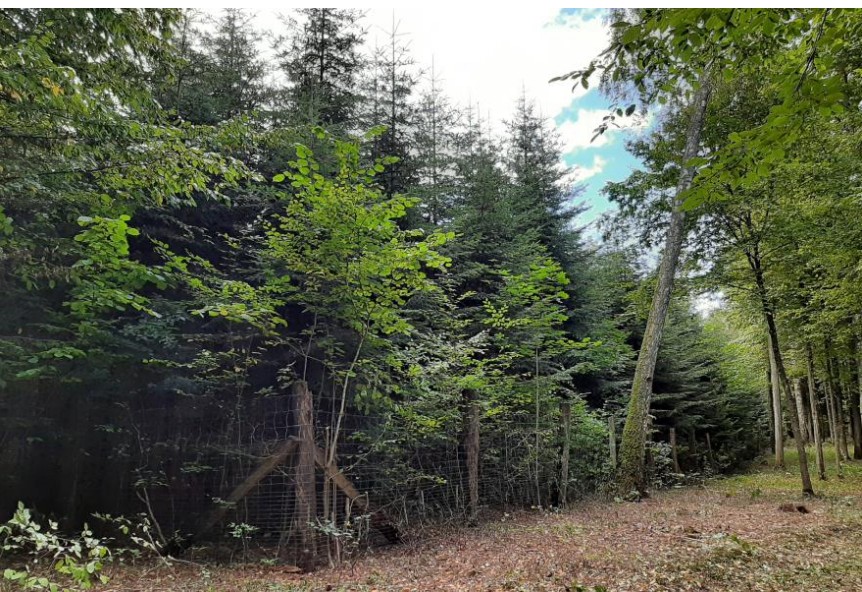

**Figure 1.** Plot No. 1 of a half-sib silver fir plantation (photo by M. Kaczyński).

After conducting research in 2018, we did not let the experimental site out of sight. In 2020 and 2021, the collection of cones was carried out. Seeds isolated from them were sown in the nursery Budy of Hajnówka FD. The pictures (Figures 2 and 3) taken from the lift used to collect the cones show that the trees are in excellent condition. In 2021, the diameter and height of the 9 tallest and thickest trees were also measured. In the same year, we made a preliminary comparative assessment of trees assigned for felling in 2015 but not harvested due to the change in the status of the object.

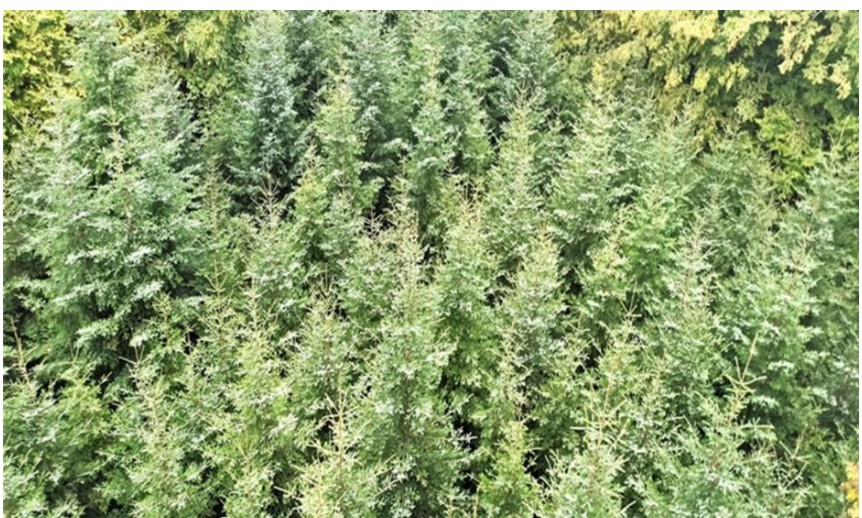

**Figure 2.** General view of plot No. 1 of the half-sib silver fir plantation from the lift when collecting cones in 2021 (photo by K. Wilamowski).

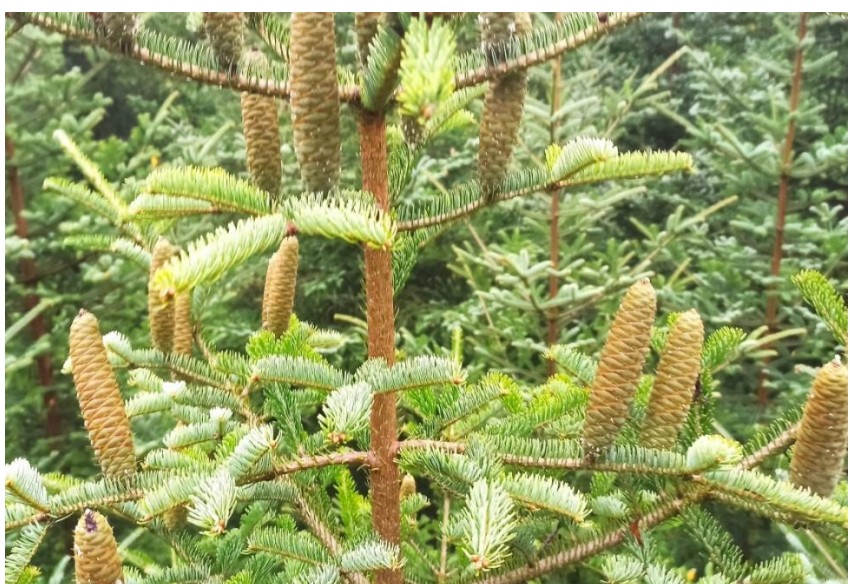

**Figure 3.** Fruiting of individual trees on plot No. 1 of the half-sib silver fir plantation in 2021 (photo by K. Wilamowski).

The crowns are pointed (positive growth trend) with dense, bright green needles. One can notice a pronounced difference in heights, which corresponds to a difference in diameters. However, the crowns of trees of the lower growth classes also look good. The aerial view shows no apparent openings in the canopy. Furthermore, 38 trees formed cones in 2021, and some showed repeated (after 2020) fruiting. As a rule, these are trees with the largest diameter. However, fruiting was also recorded on small-diameter trees. All these facts indicate that the condition of the trees on the site for three years after the study continues to be very good and is in no way a cause for concern.

### 2.2. Collection of Field Data and Its Standard Processing

Tree growth was assessed by analyzing the diameter at 1.3 m from the root collar or DBH, which represents it well and is easy to measure in dense stands (in contrast to height measurement). To determine the influence of the provenance factor for the ten half-sib families, we measured the diameters of all 701 trees to an accuracy of 0.1 cm. The representation of the families varied considerably: from 11 specimens in families 16 and 21 to 167 and 180 in 17 and 15, respectively. A relatively large number of seedlings were planted in sibs Nos. 2 and 5, respectively, 336 and 258 trees. A fairly close number of trees was initially represented by No. 3—75 pcs., No. 4—70 pcs., No. 11—66 pcs., and No. 12—38 pcs. [12].

At the time of the study, the survival of plants (as a percentage) for siblings relative to their initially planted number was No. 2—46.7, No. 3—41.3, No. 4—34.3, No. 5—28.3, No. 11—51.3, No. 12—34.2, No. 15—54.1, No. 16—61.1, No. 17—51.1, and No. 21—40.7. Despite a low survival rate, some sibs had started with such a large number of individuals that we still had many trees to evaluate. For example, sib No. 5 had the lowest survival rate—28.3. However, thanks to a high number of initial plantings—258, we have the opportunity to study 73 individuals. While sib No. 16 has the highest survival rate—61.1%, it is represented by only 11 individuals. A table with tree survival in different years of observation is presented in our previous article [12].

We wished to determine the influence of spatial (horizontal) structure on DBH in 126 randomly selected specimens (n). Accordingly, we measured the distances in four directions (north, south, west, and east) to nearest neighbors with a tape, and their average distance was calculated [12].

For our analyses, we utilized rank-based nonparametric methods to overcome distributional requirements that could not be confirmed by the data. We used the Kruskal–Wallis

(K.W.) nonparametric analysis of variance to test whether our treatment samples originated from the same distribution [27]. Before employing the K.W. test, we used the Fligner–Killeen test to confirm the assumption of equal dispersion among all the treatment levels [28]. We used Dunn's test to assess pairwise differences between treatment levels [29] and Holm's step-down method to control family-wise error rates [30]. Because the K.W. test is not one of the medians, we report the appropriate Hodges–Lehman estimators of the treatment levels [31], consistent with the K.W. test and analogous in interpretation to the other location parameters such as the median. We allowed a Type I error rate of $\alpha = 0.05$ for significance testing.

The relationship between DBH and distance to neighboring trees was modeled using a Gompertz (or Logistic) asymptotic function.

Analyses were performed using the R statistical environment version 4.0.5 [32] and R libraries PMCMRplus [33] and DescTools [34].

Identifying the influence of provenance and spatial structure factors on tree growth was based on comparing the statistical analyses of DBH (Table 1). The original data were clustered into three classes in both cases, but slightly different approaches were used to isolate them.

**Table 1.** Parameters of the DBH of the examined half-sibs.

| Half-Sib Family | Mean and Standard Deviation of the DBH, cm | Median | Lower Confidence Interval | Upper Confidence Interval |
|---|---|---|---|---|
| 4 | 7.71 (3.36) | 7.75 | 6.2 | 9.3 |
| 3 | 8.46 (3.55) | 8.5 | 7.1 | 9.9 |
| 21 | 8.52 (4.68) | 8.5 | 4.95 | 11.9 |
| 5 | 8.63 (3.50) | 8.5 | 7.65 | 9.35 |
| 16 | 8.64 (2.40) | 8.68 | 6.7 | 10.5 |
| 11 | 9.97 (3.28) | 9.8 | 8.7 | 11.05 |
| 15 | 10.05 (4.15) | 9.8 | 9.2 | 10.45 |
| 2 | 10.13 (4.04) | 10.05 | 9.4 | 10.7 |
| 12 | 11.25 (3.23) | 10.99 | 8.95 | 13.45 |
| 17 | 11.18 (4.52) | 11.1 | 10.4 | 11.8 |

### 2.3. Methodological Approach for Determining the Influence of a Provenance Factor

When determining the influence of provenance as a factor, the data were grouped into three classes based on a formal clustering procedure. The classification was based on productivity, as expressed through the average DBH of each half-sib (Table 1) based on the measure of their proximity to the average value for the entire sample.

The number of clusters (classes)—3 was established subjectively or in simplification: the average DBH is small (Class 1), medium (Class 2), and large (Class 3). The number of clusters, in this case, is optimal, and this approach is only formally subjective as it does not reduce the quality of clustering. It is widely used in practical forestry, for example, when preliminary assessing the level of fruiting, the state of natural and artificial regeneration, determining the vitality of forest stands, determining the degree of their damage by animals, insects, diseases, etc. [35–37]. Additionally, the number of objects (half-sibs) in the sample, which had to be divided into disjoint clusters, was relatively small (10), and therefore it would be inappropriate to increase the number of clusters by more than three (i.e., their fragmentation).

The solution to the clustering problem was reduced by determining for each kind the deviation of its average DBH from the average value of this parameter for the entire population (10 half-sibs), equal to 10.0 cm [12]. Half-sibs whose average DBH decreased more than 5% from the average DBH were assigned to the first class, those deviating no

more than 5% in either direction (increasing or decreasing)—to the second, and finally those whose DBH was greater than the average DBH by 5% or more—to the third (Table 2).

**Table 2.** Data clustering in the context of the provenance factor.

| Class | Half-Sibs (Deviation from the Mean Value of the DBH for the Entire Sample in %) | DBH | |
| --- | --- | --- | --- |
| | | Mean and Standard Deviation, cm | Coefficient of Variation,% |
| 1 | 4 (−22.9), 3 (−15.4), 21 (−14.8), 5 (−13.7), 16 (−13.6) | 8.44 (3.49) | 41 |
| 2 | 11(−0.3), 15 (+0.5), 2 (+1.3) | 10.08 (4.02) | 40 |
| 3 | 12 (+12.5), 17 (+11.8) | 11.19 (4.43) | 37 |

As you can see, the deviation of the DBH of the species presented in the central, second class, is insignificant and amounts, as we initially assumed, no more than 5% of the average diameter for the entire population (701 trees), while for the first and third classes these deviations ("−" and "+") more than 10%, sometimes much more (Table 2). These data, however, are only preliminary evidence of the legitimacy of clustering, i.e., there is a need to confirm it using better statistics. At the same time, it will answer the question of the influence of provenance on DBH.

### 2.4. Methodological Approach for Determining the Influence of the Factor of Spatial Structure

In our argument concerning spatial structure, we work with the concept of "growth area" [38,39]. It corresponds to the "feeding area" [40,41] and conditionally implies the space available to a particular tree and not used by the neighboring trees.

When determining the influence of the spatial structure (defined as the average distance to the nearest four trees in the directions of the cardinal points), the data on DBH were also divided into three classes. That was done for better comparison with the provenance factor. We used the data obtained earlier [12] and presented it in Table 3. Moreover, the partitioning of a set of objects (126 randomly selected trees), in contrast to clustering by provenance factor, was more consistent with the understanding of automatic classification, which excluded any subjectivity.

**Table 3.** Basic statistical indicators are used to group data by class in determining the influence of the factor of spatial structure [12].

| Indicator | Mean and Standard Deviation | Minimum Value | Maximum Value | Coefficient of Variation, % |
| --- | --- | --- | --- | --- |
| Ln [a] | 2.13 (1.09) | 0.77 | 6.15 | 52 |
| Ls [b] | 1.94 (0.88) | 0.69 | 6.30 | 46 |
| Le [c] | 1.89 (0.72) | 0.58 | 3.75 | 38 |
| Lw [d] | 1.81 (0.73) | 0.59 | 4.26 | 41 |
| Lav [e] | 1.93 (0.49) | 1.10 | 3.42 | 26 |

[a]—distance to the nearest tree in a northerly direction (distance in a row), m; [b]—distance to the nearest tree in a southerly direction (distance in a row), m; [c]—distance to the nearest tree in the east (distance between rows), m; [d]—distance to the nearest tree in the west direction (distance between rows), m; [e]—average distance to the nearest trees, m.

The following mathematical manipulation was applied. The minimum value was subtracted from the maximum value of the mean distance between trees, and class boundaries were established by dividing the difference by three. It was not difficult to establish the

middle of the class interval. Additionally, the actual average distance between trees in each class was determined (Table 4).

**Table 4.** Data clustering in the context of the spatial structure factor.

| Class | Distance | | | | DBH | |
| | Class Boundaries | The Middle of the Class Interval | Mean and Standard Deviation, m | Coefficient of Variation, % | Mean and Standard Deviation, cm | Coefficient of Variation, % |
|---|---|---|---|---|---|---|
| 1 | 1.10–1.87 | 1.49 | 1.59 (0.19) | 12 | 9.6 (4.1) | 43 |
| 2 | 1.88–2.65 | 2.26 | 2.18 (0.20) | 9 | 13.4 (4.9) | 37 |
| 3 | 2.66–3.42 | 3.04 | 3.07 (0.21) | 7 | 16.9 (4.2) | 25 |

Note that the last two parameters, which have a similar meaning but are determined in different ways (empirically and based on statistical processing of actual data), practically do not differ. This testifies to the legitimacy of the methodological approach used in the cluster classification of objects in studying the significance of the role of the spatial structure. Next, the statistical parameters DBH were determined for each class (Table 4). Thus, dividing the sample into three classes and grouping each class tree with a similar spatial structure (average distance to neighbors) allowed us to identify its effect on productivity (DBH) and compare it with the influence of the provenance factor.

### 3. Results

#### 3.1. The Influence of the Provenance Factor

Dunn's test of three classes of DBH by provenance factor showed a statistically significant difference between the DBH of trees of all three classes (Table 5), even in comparable pairs of classes with a close DBH, for example, Class 1/Class 2 and Class 2/Class 3.

**Table 5.** The *p*-value result of pairwise comparisons of classes by provenance (for $\alpha = 0.05$).

| Comparison Groups | *p*-Value |
|---|---|
| Class 1/Class 2 | 0.00015 |
| Class 1/Class 3 | 0.0000000072 |
| Class 2/Class 3 | 0.00227 |

All three classes have a wide range of values. Class 2 is represented by the largest sample (371 trees), the mean value of the DBH, which is 10.08 cm (Table 6). However, the smaller samples, represented by Class 1 (150 trees) and Class 3 (180 trees), are still sufficient to obtain representative data [42].

**Table 6.** The volume of classes of the provenance factor and the amplitude of indicators of DBH in them.

| Class | N, pcs | Mode, cm | Hodges-Lehman Estimator | Lower Confidence Interval | Upper Confidence Interval |
|---|---|---|---|---|---|
| 1 | 150 | multi * | 8.4 | 7.81 | 8.99 |
| 2 | 371 | 9 | 9.9 | 9.49 | 10.35 |
| 3 | 180 | 11 | 11.1 | 10.45 | 11.75 |

* distribution having several modes.

#### 3.2. The Influence of the Spatial Structure

The analyzed feature also depends on the spatial structure, which the Kruskal–Wallis test confirmed. However, Dunn's test for pairwise differences between all pairs revealed

that only Class 1 compared to Class 2 (Class 1/Class2) and Class1/Class3 differ significantly (Table 7).

**Table 7.** The *p*-value result of pairwise comparisons of classes by spatial structure (for $\alpha = 0.05$).

| Comparison Groups | *p*-Value |
|---|---|
| Class 1/Class 2 | 0.000338039927 |
| Class 1/Class 3 | 0.000000019719 |
| Class 2/Class 3 | 0.12146648694 |

Class 1 is represented by the largest number of trees (68 trees), with the smallest average DBH. Class 2 has the greatest range in DBH. Classes 2 and 3, despite the difference in mean DBH (Table 8), do not differ statistically (Table 7). The range of variation in class 3 is the smallest.

**Table 8.** The volume of classes of the spatial structure factor and the amplitude of indicators of DBH in them.

| Class | N, pcs | Mode, cm | Hodges-Lehman Estimator | Lower Confidence Interval | Upper Confidence Interval |
|---|---|---|---|---|---|
| 1 | 68 | 8.5 | 9.3 | 8.29 | 10.35 |
| 2 | 45 | 23 | 13.4 | 11.75 | 14.9 |
| 3 | 13 | multi * | 17.0 | 14.55 | 19.65 |

* distribution having several modes.

The influence of the spatial features is superimposed on the phenotypic manifestation of genotypes (average DBH) characterizing certain half-sibs. We performed a correlation analysis (Table 9) using the tree DBH and spatial distribution of the neighboring trees (Tables 1 and 3).

**Table 9.** The relationship between DBH and the distance to the nearest trees in the directions of the cardinal points.

| Indicators * | Ln | Ls | Le | Lw | Lav |
|---|---|---|---|---|---|
| Coefficient of correlation | +0.65 | +0.69 | +0.45 | +0.67 | +0.72 |

* See Table 3.

The relationship between the analyzed indicators is weak if the correlation coefficient is less than $\pm 0.3$; from $\pm 0.3$ to $\pm 0.7$—the relationship is the moderate strength; if the correlation coefficient is greater than $\pm 0.7$—the relationship is strong [42]. In our case, the relationship between the average stem DBH and the average distance to the nearest trees is strongly positive. Moreover, with the distances in specific directions, it has a moderate positive strength (Table 9).

*3.3. Comparison of the Influence of Provenance Factor and Spatial Structure*

We examined the differences between the DBH classes and the breadth of the variation within each class (Figures 4 and 5). Using these data, we can make a preliminary conclusion about the presence (or absence) of dependence of DBH variation on factors of provenance and spatial structure and compare their influence on DBH.

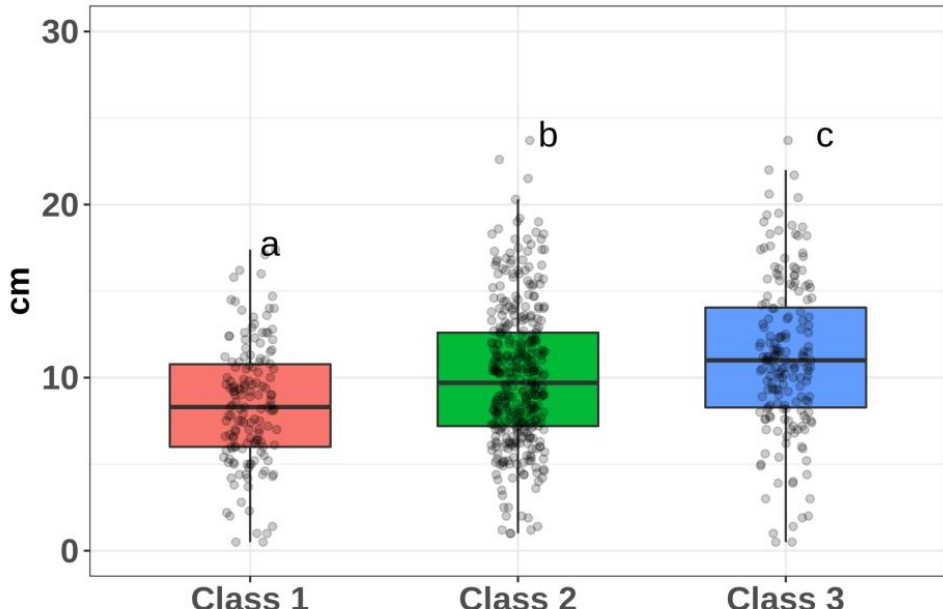

**Figure 4.** The relationship between tree DBH and the provenance factor. The lower and upper hinges correspond to the first and third quartiles (the 25th and 75th percentiles), and the centerline corresponds to the median. The upper whisker extends from the upper hinge to the largest value no further than 1.5*IQR from the hinge (where IQR is the interquartile range or distance between the first and third quartiles). Likewise, the lower whisker extends from the lower hinge to the smallest value no further than 1.5*IQR from the hinge. The different lower case letters (a, b, c) above the means of the treatments (Class 1, 2, 3) are significantly different at $p$ = 0.05. Points on the plot were offset to avoid overplotting of identical values.

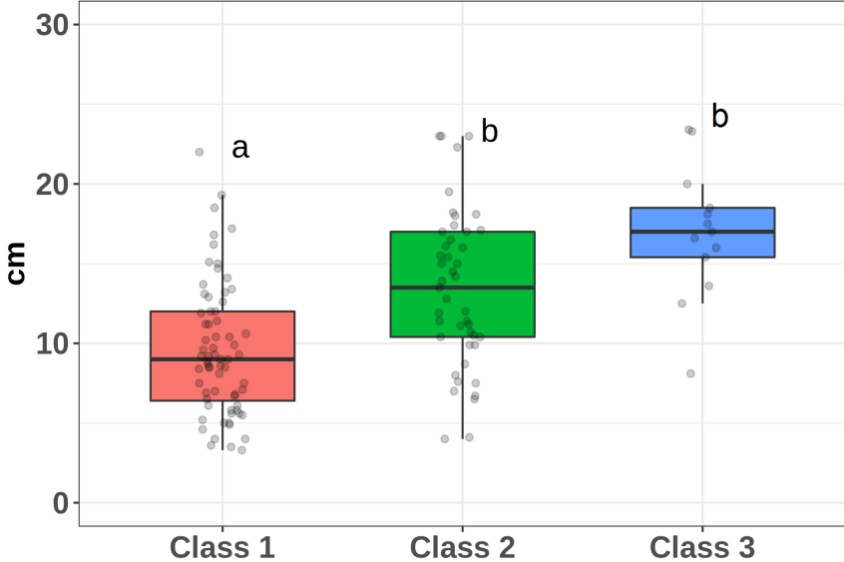

**Figure 5.** Dependence of the DBH on the spatial structure factor. The lower and upper hinges correspond to the first and third quartiles (the 25th and 75th percentiles), and the centerline corresponds to the median. The upper whisker extends from the upper hinge to the largest value no further than 1.5*IQR from the hinge (where IQR is the interquartile range or distance between the first and third quartiles). Likewise, the lower whisker extends from the lower hinge to the smallest value no further than 1.5*IQR from the hinge. The different lower case letters (a, b) above the means of the treatments (Class 1, 2, 3) are significantly different at $p$ = 0.05. Points on the plot were offset to avoid overplotting of identical values.

### 3.4. Most Productive Phenotypes

Nine trees ranging from 20.3 to 23.7 cm in DBH and 18.0 to 22.4 m were identified (Table 10). While predictions of near-term growth can be challenging, it seems to us that it is unlikely to slow down. If we compare the estimated features of these 29-year-old trees with the tabular ones [43], we will see that they correspond to 55–60-year-old natural stands in diameter and 50–60 years old in height, growing in conditions of the 1st class of stand quality.

**Table 10.** Characteristic of the best trees (2021).

| Row | Tree Number | Half-Sib | DBH, cm | Tree Height, m | Average Distance to the Nearest Trees, m |
|-----|-------------|----------|---------|----------------|------------------------------------------|
| 14 | 20 | 2 | 21.5 | 18.0 | 3.05 |
| 21 | 20 | 15 | 22.6 | 20.1 | 2.13 |
| 22 | 16 | 17 | 22.0 | 19.2 | 1.89 |
| 28 | 24 | 2 | 20.3 | 21.2 | 3.14 |
| 39 | 17 | 15 | 23.7 | 20.9 | 2.76 |
| 42 | 15 | 17 | 21.0 | 20.5 | 3.85 |
| 50 | 9 | 17 | 20.6 | 21.8 | 2.50 |
| 50 | 13 | 17 | 23.7 | 22.4 | 3.24 |
| 50 | 16 | 17 | 20.4 | 20.3 | 2.93 |

Half (55.5%) of the most productive phenotypes belonged to half-sib No. 17. Interestingly, the energy of the best trees is enough not only for intensive growth but also for frequent fruiting—some of them have recorded the formation of cones for two consecutive years. However, among these nine trees, not one is in the 1st class in terms of the spatial structure factor (Table 4).

### 4. Discussion

We note that the established significant variability of average DBH of half-sibs (Table 1) occurs in the absence of differences in edapho-climatic conditions, characteristics of the forest cultivated area, age, and type of planting material, timing, method of establishment, and maintenance of the plantation. The provenance factor's influence is reflected by the membership of trees in a particular half-sib family. However, within each half-sib, a certain amount of phenotypic variability occurs. This variability might be due, among other things, to possible cross-fertilization with the influence of paternal components, which may differ genetically. Moreover, as noted above (see section "Introduction," [19]), trees not only from Tisovik but also from stands of unknown origin in the Polish part of the Forest may participate in pollination. The closest location (subcompartment 453Aa) is 8.5 km from the refugium (Figure 6). If this is the case, then given the growing aridity of the climate, an increase in genetic mixing is regarded as a positive factor. However, the opposite process is also possible—with pollen from Tisovik pollinating the macrostrobils on trees in sub-compartment 453Aa.

As shown above, the parameters of phenotypic variability differ in different half-siblings, which may evolve into actual genetic differences. This potential for future genetic variability suggests the need for maximum conservation of each tree—a potential source of heritable valuable economic traits—in thinning.

Note that the biological features of fir fully contribute to this outcome. On the one hand, due to the pronounced taproot system of fir, the growing space of individual trees is relatively small. On the other hand, this species is characterized by exceptional shade tolerance [44–46]. These results suggest that managers reduce harvest impacts of future silvicultural and increase genetic opportunities by minimizing felling intensity at the

cleaning stage. For example, in 2015, the marked tree in Figure 7 was initially designated for felling because of its reduced growth compared to the neighboring trees. However, since this experimental site was assigned to the UNESCO protected zone II, where any felling is prohibited, this tree was preserved. This "reprieve" was borne out by its subsequent performance similar to neighboring trees by 2021.

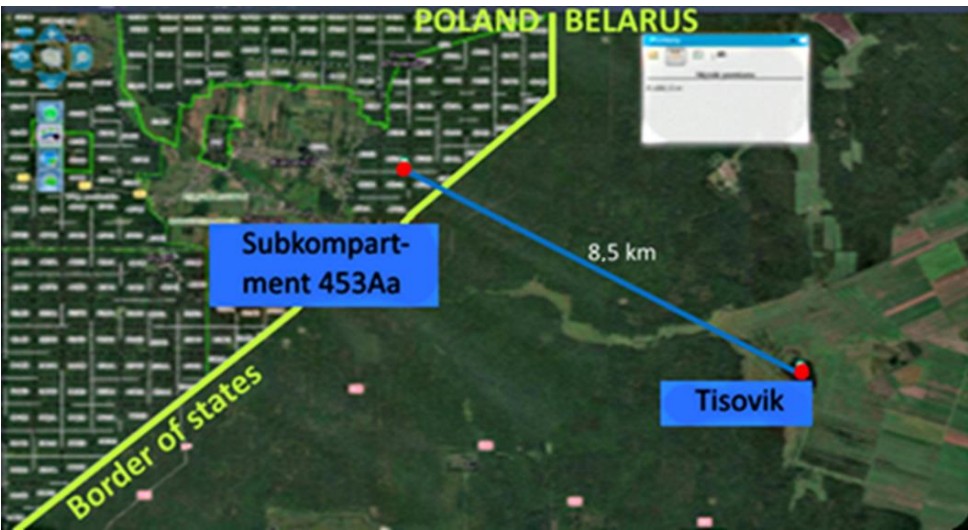

**Figure 6.** Localization of the silver fir refugium and the nearest fir stand in the Polish part of the Białowieża Forest (photo by W. Klimiuk).

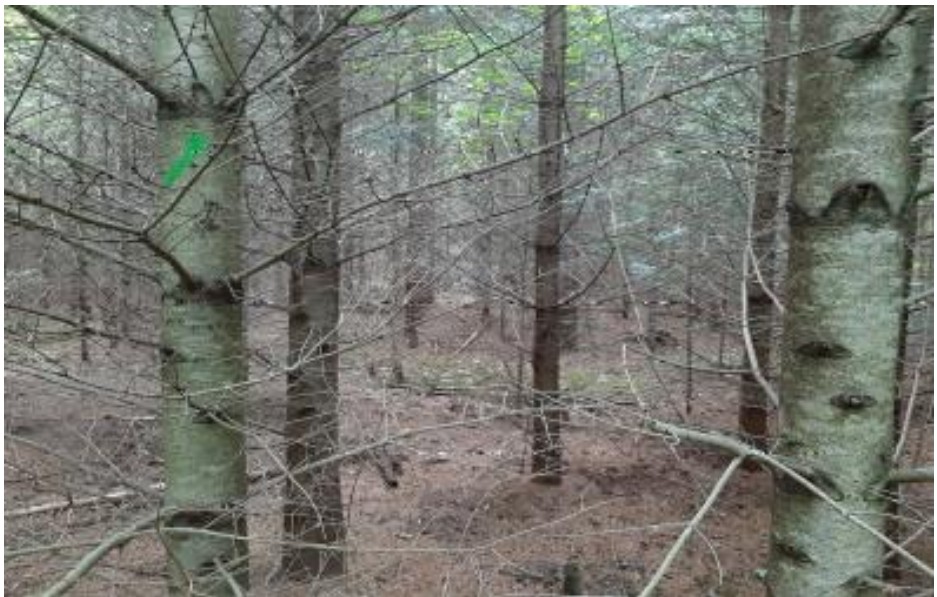

**Figure 7.** Silver fir, which was lagging in growth in 2015 (it was then marked in green, which means an initial assignment for harvest, but not felled). Compare this fir in 2021 to neighboring trees that were not assigned for felling 6 years ago (photo by A. Marozau).

In this regard, we also note that, according to Goncharenko [47], the fir in Tisovik "despite its isolation and small size, has not yet experienced the influence of inbreeding, and although only 20 trees were preserved, as a result of crosses (cross-pollination) between them, one can obtain 13,122 genetically different offspring".

Nevertheless, the issue of the manifestation of inbreeding in the Tisovik tract should be approached more carefully and, perhaps, not so optimistically [48]. Because the population, represented by only about 20 trees [7,12,47], is very small, and they live in a very small

area. In addition, perhaps it is the effect of inbreeding that can explain the lowest survival rate in family No. 5—28.3 (only 73 trees out of 258 survived) [12], as well as the significant differences in diameter both between sibs and within sibs, as can be seen from our data. Although, in the latter case, the effect of inbreeding, even if it takes place, is manifested against the background of the natural process of biological differentiation of trees (Kraft classes or social classes of trees position in canopy [48]).

Moreover, for an exceptionally shade-tolerant fir, it is characteristic that not only trees of IV, but even sometimes Va of Kraft classes are not doomed to fail. Most of them are not in a state of gradual mortality. On the contrary, their condition corresponds to a live plant, only with very slow growth rates. In addition, this is precisely why the statistical analysis showed a large standard deviation and, accordingly, the coefficient of variation of DBH (Tables 1, 2 and 4). In natural stands, when the dominant trees approach the climax stage, it is the representatives of the lower tiers that begin to take over their role [44–46].

To obtain a genetically richer silver fir population originating from the seeds of the planned vegetative seed plantation created by grafting, we must first select the best phenotypes from all sibs.

Therefore, when creating it, you should not focus only on the top nine trees identified at the moment. According to Barzdajn [49], who was involved in the restitution of silver fir in Sudety, clones should number at least 100. This conclusion implies that phenotypic studies must be repeated periodically to identify the best trees (Table 10). In our opinion, a frequency of 5 years will be necessary. These remeasurements will make it possible to record the emergence of new candidates for superior trees since, as we noted earlier, there is a pronounced positive dynamics of tree growth [12] (Figures 2 and 3). Additionally, two more plots are awaiting their turn at this object. The most critical condition for the success of future vegetative seed plantations is the maximum representation of all the best phenotypes from all half-sibs families.

The closest positive correlation of DBH occurs with the distance to the nearest neighboring tree in the southern direction (0.69) and the lowest (0.45) in the eastern direction (Table 9). These results suggest that the creation of conditions for an increase in illumination in the stand without excessive felling of trees (i.e., an increase in the distance to a neighboring tree in the southern direction with the highest intensity of solar radiation) has a positive effect on the growth of trees (an increase in their DBH). As it turns out, *A. alba*, which is a classically shade-tolerant species, responds positively to improving light conditions under these conditions, even at a young age.

The relationship between the DBH and the average distance to neighboring trees (Figure 8) suggests a certain "threshold value" of the average distance between trees in a stand of a given age. Beyond this threshold, further thinning will not lead to an increase in the productivity of the stand nor an improvement in its general condition. At the moment, this threshold indicator is 2.3 m (Figure 8). However, this distance is not permanent and will change with the age of the stand.

Thus, 2.3 m is a "threshold," which we believe determines the end of the influence of spatial structure on DBH increase. This factor ceases to be limiting [50,51]. Even though the curve of the graph continues to move very slightly upward, from a silvicultural point of view, a further increase in the average distance will lead to an unnecessary decrease in the density of the stand, a reduction in its overall growth and volume, and underutilization of forest area by root systems of trees. Furthermore, an excessive increase in the sparseness of the stand negatively affects the light regime of the shade-tolerant fir. A simple calculation shows that with an increase in the average distance between trees from 2.3 m to 2.5 m (i.e., when the red cross in Figure 8 moves to the right), it is necessary to additionally remove about 300 trees from a hectare, which will lead to a decrease in stand density by 0.1 units.

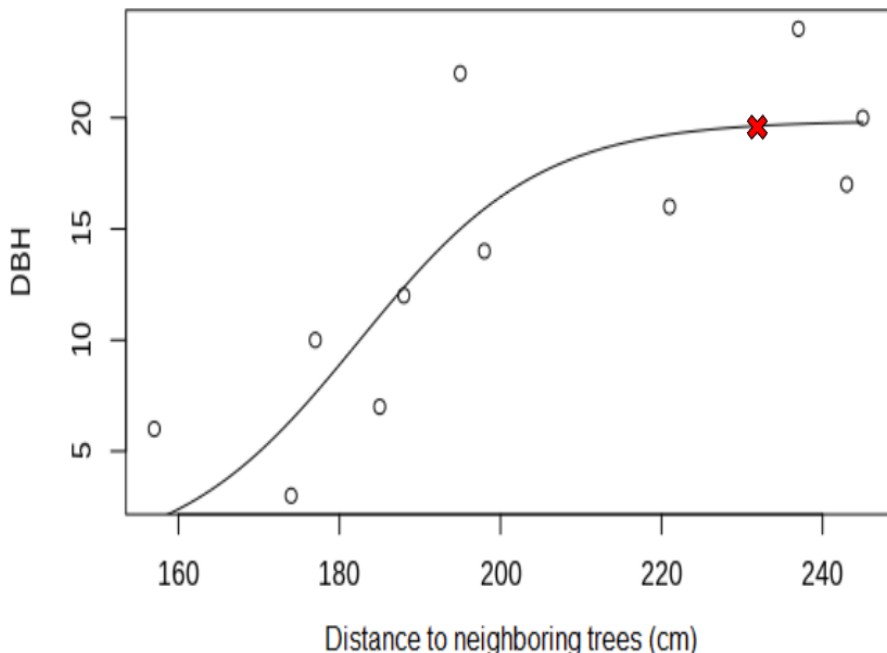

**Figure 8.** Dependence of the DBH on the average distance to neighboring trees. The line was fitted using a Logistic (or Gompertz) nonlinear relationship that allows for estimation of the asymptote of 2.3 m (lower 95% bound, upper 95% bound).

The same cautious approach to tree cutting should be followed in the future. Of course, the parameters of the spatial structure will change, and this is the subject of future research. However, the principles of silvicultural care remain as follows: (a) only obvious outsiders are cut down (trees at the left-hand side of the distribution curve), and (b) tree stand care is carried out using more frequent, low-intensity entries rather than infrequent intensive thinnings.

In this case, the presence of individuals with an increased DBH and indicators of spatial structure exceeding the "threshold" can only be interpreted as a consequence of the influence of a provenance factor. As shown in Table 10, in seven out of nine trees (78%) with the largest DBH (candidates for plus trees), the average distance to neighbors exceeds 2.3 m. A genetic study of these trees, planned for the future, will allow a more substantiated statement on the validity of this assumption.

Following the methodological approach of Rogozin [41], we use half the average distance between trees (1.15 m) as the radius of a sufficient "feeding area" [40,41] or "growth area" [38,39]. As a result, we get that the growth area of each tree is 4.15 m$^2$, and the density of the stand is 2410 pcs/ha (where it is 3186 pcs/ha at present). It is necessary to approach this indicator, as already mentioned, by selecting the weak intensity according to the grass-root method, removing especially the trees of the 2 and 4 cm diameter classes. In this way: (1) the biology of the species (shade tolerance) is taken into account, (2) the vertical structure is formed, and (3) the gene pool of *A. alba* is preserved.

In a situation where a pronounced differentiation is currently taking place in the forest stand (Table 1), the future of the dominant trees, especially advantage tree candidates, is of interest. Can their growth be improved? It was found that there is an unequal dependence of the change in DBH on the distance to the nearest tree in the direction of the different world sides (Figure 9).

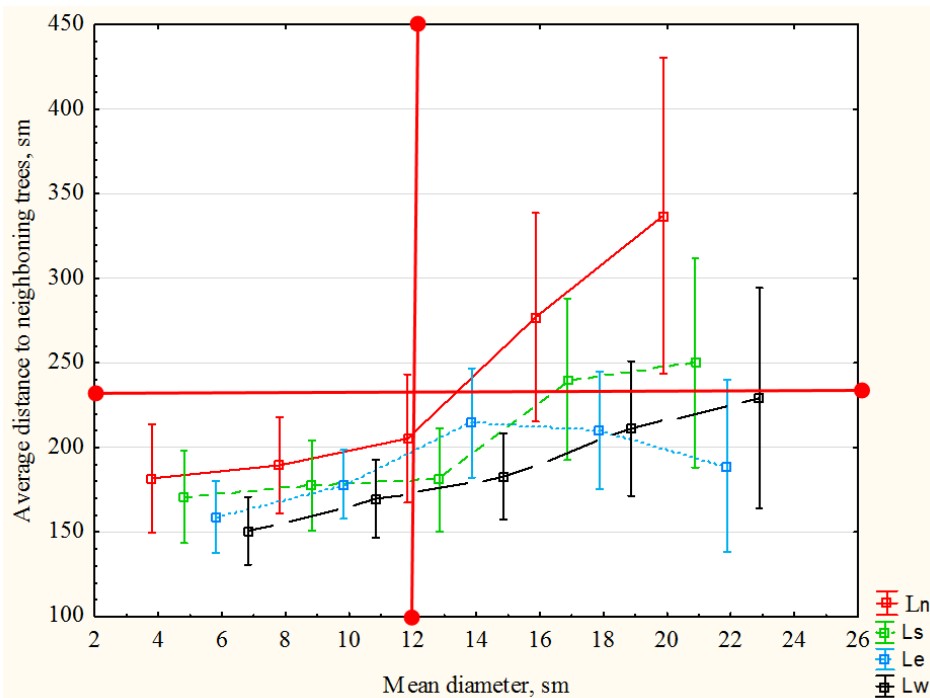

**Figure 9.** Dependence of DBH on the distance to neighboring trees according to the geographical orientation.

The starting point for evaluating this dependence is the intersection of the two perpendiculars on the axis of this diagram. The vertical perpendicular corresponds to a DBH of 12 cm, representing the diameter class above the average stand DBH (9.97 cm, [12]). The horizontal line corresponds to the "threshold" of the spatial structure (see above).

The thinning required to increase the DBH, expressed as the distance to the nearest tree in the various directions, takes the following form (in decreasing order): north, south, west, and east. That is, by focusing on the removal of trees in the rows (north-south direction) and at the same time cutting less intensively in the direction between the rows (west-east), it is possible to influence the increase in DBH of the dominant trees. This opportunity arises from the trees' initial very dense arrangement in the rows (1 m, and sometimes less, Table 3).

The complexity and ambiguity determining the influence of the spatial structure are illustrated by this analysis of the randomly located sample (Figure 10).

As can be seen, the planting lines are still visible. However, the initially regular (systematic) placement of trees according to the 1.3 × 1 m design becomes uneven. Biogroups of trees (highlighted in green) and gaps in the forest stand (highlighted in orange) begin to form. Note that from a bird's eye view, the unevenness of the canopy of the crowns, due to their development, is not yet so apparent (Figure 2). What may be the cause of the uneven spatial structure of the stand? First of all, we note the possible accretion of fir roots [52] and the formation, thus, of microgroups, each of which has a common root system. Some plants died due to late spring frosts in the early years (see Korczyk [53]). Perhaps the negative influence of excessive solar radiation was also uneven in different parts of the plot. In addition, this non-uniformity in the distribution of plants persisted subsequently (we did not find any information in the literature about the addition of crops). After thinning, it may have even increased somewhat under the influence of the subjective factor (selection of trees for cleanings). However, the root causes of this phenomenon (uneven distribution of trees) are not yet known [41]. Therefore, we can only note the appearance of fundamentally new attributes of the morphogenesis of the stand: biogroups (microcenoses) and gaps in the stand.

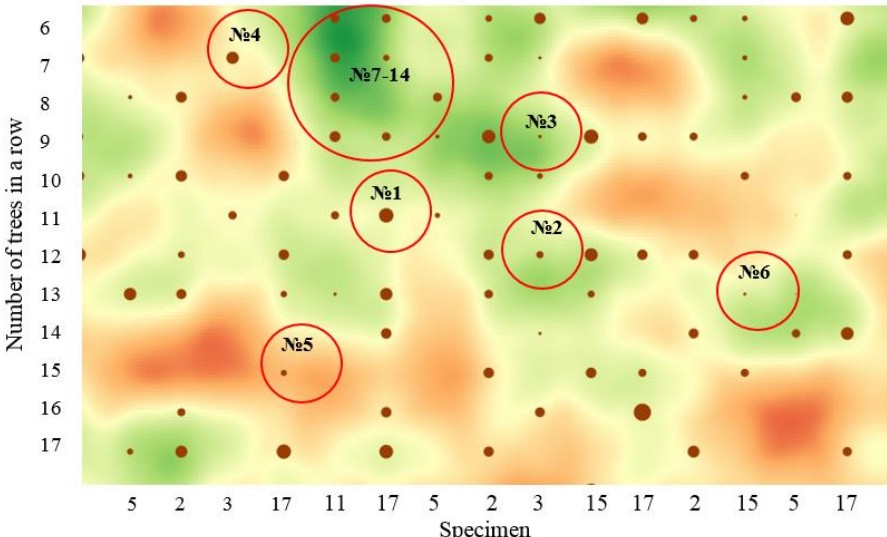

**Figure 10.** Arrangement of trees placement in an arbitrarily selected stand's fragment (differences in the DBH are reflected by different sizes of circles). Drawing by A. Kaliszuk.

From the analysis of the fragment of the spatial model (Figure 10), it follows that the trees with neighbors that are behind in growth or have a greater distance to them also have a larger DBH (Nos. 1, 4). Conversely, the DBH of the focus trees decreases with decreasing distance to neighboring trees or an increase in the neighboring trees' DBH (Nos. 2, 3). These two situations are well explained by the weakening and strengthening of intraspecific competition. However, then, do we explain the same DBH of the trees Nos. 7–14 growing in the biogroup? Perhaps there is already the partnership effect (group effect) [41] (see Figure 11). In addition, in this context—perhaps there is no need to strive for an even distribution of trees [54]? In nature, *A. alba* is usually found in an admixture of the main tree species in groups because it has relatively heavy seeds [46]. In addition, as it turns out, it grows better with artificial reproduction in groups; in particular, it is highly resistant to biotic and abiotic factors [55].

And what influenced the insignificant DBH of the trees Nos. 5 and 6 growing in the gaps? After all, the competition factor is not the limiting influence at present. Perhaps they were suppressed before the late cleaning and still cannot rebuild? Or is this perhaps the result of genetic differentiation? Approximately the same questions arise in analyzing any other fragment of a half-sib plantation.

It must be admitted that the above analysis is objectively a simplified approach that does not take into account other complex influences that are challenging to determine and not consistently observable by managers and scientists (allelopathy, root competition, local specificity of the soil, the state of the plants before planting and late cleaning, etc.). According to Rogozin [41], the proportion of unknown factors that determine the size of trees in old pine stands can reach 60%.

When assessing the influence of the provenance factor on the diameter, it was shown that all three classes are statistically significantly different from each other in terms of the p-level of significance. In addition, there is a pronounced dependence of the DBH on the provenance factor (Figure 4).

A different situation occurs when assessing the influence of spatial structure. While classes 1 and 2, as well as 1 and 3, differ significantly from each other ($p < 0.01$), the difference between classes 2 and 3 is not significantly significant ($p = 0.12$). Nevertheless, we note that the tendency to increase in diameter from class 2 to class 3 is visually apparent (Figure 5).

In light of the above, there is a reason to once again turn to the conclusion illustrated in Figure 8. The distance between the trees (the density of the stand), which provides them with favorable conditions at a given age is 2.3 m, the middle of the class inter-

val No. 2 (Table 4). The silvicultural interpretation of the statistically insignificant difference between classes 2 and 3 is the same: a borderline acceptable distance between trees—"threshold"—exists. We found the same result using two different methods, supporting our conclusions.

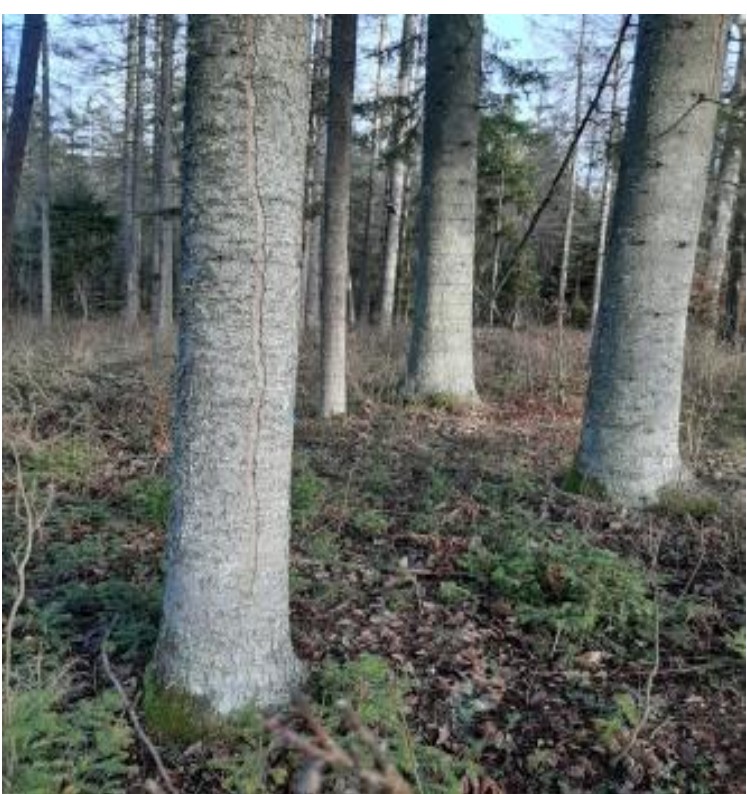

**Figure 11.** Biogroup of old firs in the Polish part of Białowieża FD (subdistrict Grudki, subcompartment 498 Ci) (photo by A. Marozau).

## 5. Conclusions

The study results give reason to believe that at this particular age, provenance has a more significant influence on the growth of the trees in DBH throughout the whole range of its variation than has the spatial structure. However, it was also found that the importance of the last factor should not be discounted, especially in the range of small and medium distances between trees (classes 1 and 2), where it can determine their parameters.

Free pollination (father unknown) of a particular mother tree in the Tisovik tract is the basis for the emergence of the half-sibs we studied. In this respect, each of the trees on the plantation of European silver fir, even within the same family, is genetically differentiated and may differ in characteristics to one degree or another. Therefore, to maximize future silvicultural and genetic opportunities, we should preserve as many trees as possible.

The nine best trees were identified, which are the prototypes of plus trees. Their seeds and shoots can already find a practical use for the early creation of a seed base of autochthonous silver fir. By starting this work now, in an experimental stand of about 30 years, we will accelerate the emergence of seed plantations by at least 50 years.

Family No. 17 is of particular interest for further genetic study. A significant number of specimens were identified in this half-sib, with DBH significantly exceeding the average of the plantation. However, this does not mean we should concentrate only on this half-sib. On the contrary, all semi-sibs should be maximally represented in the seed base in order to maintain or even enhance the genetic diversity of future autochthonous silver fir forests.

The actual planting density at the time of the study was 3186 plants/ha, while the threshold value determined by us is 2410 plants/ha. This particular hold can be used in practical forestry when thinning silver fir trees at about 25–30 years. The stand should be



"brought" to the threshold value gradually, performing regular fellings of low intensity and removing only obvious outliers.

In light of the catastrophic situation with Norway spruce in Białowieża Forest, studies on various aspects of growing local silver fir as an alternative are of undoubted scientific and practical value.

**Author Contributions:** Conceptualization, A.M., T.O. and U.K.; methodology, A.M., U.K., D.B., J.N. and T.O.; software, U.K. and D.B.; validation, D.B., J.N., T.O., W.K.M. and T.H.; formal analysis, A.M., U.K., D.B., J.N. and T.O.; investigation, A.M., T.O. and J.N.; resources, A.M. and T.O.; writing—original draft preparation, A.M., U.K., T.O. and J.N.; editing, T.O., J.N., D.B., W.K.M. and T.H.; supervision, A.M., D.B., J.N. and T.O.; project administration, A.M. and T.O.; funding acquisition, A.M. All authors have read and agreed to the published version of the manuscript.

**Funding:** The study was carried out within the framework: WZ/WB-INL/2/2021 and financed from the science funds from the Ministry of Science and Higher Education in Poland.

**Data Availability Statement:** No publicly archived dataset.

**Acknowledgments:** The authors thank student Adam Kaliszuk for participation in fieldwork and primary data processing. Some material from the engineering thesis of Adam Kaliszuk "Analysis of the breeding status of a 23-year-old ancestral cultivation of silver fir (*Abies alba* Mill.) in the Hajnówka Forest District" at Bialystok University of Technology, Poland, was referenced in the article with full consent of the main author.

**Conflicts of Interest:** The authors declare no conflict of interest. The funders had no role in the study's design, in the collection, analyses, or interpretation of data, in the writing of the manuscript, or in the decision to publish the results.

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
