# Peer review of "The Influence of the Provenance and Spatial Structure on the Growth of European Silver Fir (Abies alba Mill.) of Autochthonous Origin in a Forest Plantation in the Białowieża Forest"

_forests, doi:10.3390/f13060831_

Round 1
Reviewer 1 Report
The paper is well revised for publication.
Reviewer 2 Report
I have read the replies of the authors and find now that the MS has been substantially improved, and it is acceptable for publication in Forests.
This manuscript is a resubmission of an earlier submission. The following is a list of the peer review reports and author responses from that submission.
Round 1
Reviewer 1 Report
The authors' manuscript "The Influence of the Genetic Factor and Spatial Structure on the Growth of European Silver Fir (Abies alba Mill.) of Autoch-thonous Origin in a Forest Plantation in the Białowieża Forest " clarified the two major factor for growth of European Silver Fir (Abies alba Mill.) in the Białowieża Forest. The author compared the 10 half-sib families diameter to show the influence of genetic factors and spatial structure. And founded that the genetic factor's influence at the pole wood stage manifests itself more clearly than spatial structure. The results in this manuscript will play an important role in further research on the factor of European Silver Fir (Abies alba Mill.) growth . It is worthy to publish this manuscript in the journal of "Forests". However, this manuscript is still exist some problem. It needs a moderate revision before publication. Therefore, my specific comments are presented below.
The language in places needs to be improved; please have it revised by a professional science editor;
For the Abstract:
The author mainly focused on the characteristics and growth status of silver fir. However, the results, aims and conclusion has been ignored in Abstract part.
For Introduction
There is repetition in the second paragraph
The objectives of this study were evaluate the variability of the most important silvicultural parameter-diameter-and thus reveal the degree of possible differentiation both between the families of the studied silver fir half-sibs and within families, however, it is not detailed in the introduction.
For the Materials and methods
1)The format of the table in the manuscript is in correct, it should be a three-line table.
2) 2.3. This part had divided 10 half-sib family into three class, the author should be present in the table. And the words shouldn’t repeat the data which had showed in the table.
3) The part should be reorganizing logic. Most of the content is meaningless.
For the Results
Table 5 The results of Pairwise comparisons of сlasses should be more comprehensively.
Table 10 should be a part of Materials and methods
The analysis method the author use is very simple and lack of innovation which can explain the aims.
In results, what is the approaches of effective regulation of the spatial structure of the stand in the near term?
For the Discussion
The figure is not clearly.
Lack of relevant references to support the author results.
For the Conclusion
At the conclusion part, the author don’t demonstrate the practical meaning of this paper.
Reviewer 2 Report
Manuscript reviewed:
Forests 1425261
“The Influence of the Genetic Factor and Spatial Structure on the Growth of European Silver Fir (Abies alba Mill.) of Autochthonous Origin in a Forest Plantation in the Białowieża Forest”
Basic report and major comments - suggestions for improving the manuscript
The manuscript is a one-year study on the trees’ growth (based on the diameter of trees in the plantation, at the age of 26/(4+22) of half-sib descendants of silver fir. Even if the work seems simplistic, it can be of interest due to the approach, case study and originality of the biological material. The context is also of interest and unfortunately quite generalized, due to the disappearance of an extremely important forest species such as silver fir, and in the study area including the spruce, strongly affected by harmful insects (Ips).
The manuscript is topically and interesting research and fits the purpose of the journal. Even if the subject of the paper is quite specialized (as a niche), it could be of interest to a large audience. The disadvantages of the manuscript consist in an insufficient presentation, description and capitalization of the research data. Thus, the results presented may seem too preliminary or superficial, starting with the abstract and the full text, or the work unfinished or superficial due to some inaccuracies. Many aspects of importance for the understanding of the manuscript by the readers, of the material and the working method, and even of the results are not offered in the presentation. They are just 'covered' by citations-references to other works (clearly, many of them performed on the same subject). In this way, the reader will not properly understand the results, discussions, sentences and hypotheses formulated, and may lose interest in this research.
In addition, although DBH is the most important element of tree growth in forestry, it is a pity that the authors did not present other elements of tree growth or their habitus (trees height, architectural shape etc.), vegetation or health etc., even if the density of the trees was an impediment, as they said. These would have allowed obtaining more consistent data, especially since the study refers to the genetic factor, spatial structure and interactions. Probably, the use of a drone would have helped to obtain additional data and with a better possibility to correlate the factors and to interpret the connections.
Update: Checking the resources from ms references, I just found a manuscript regarding the similar subject and forest stand (with many things in common), published in Baltic Forestry, which states, from the abstract: "The trees showed no signs of disease or insect damage, their crowns were conical in shape and the shoots were densely covered with needles.", see ‘Reintroduction of the European silver fir (Abies alba Mill.) in Białowieża Forest’, Baltic Forestry, Vol 27 No 1 (2021):
https://www.balticforestry.mi.lt/ojs/index.php/BF/article/view/527
In addition, much of the data in that paper would be useful in this manuscript, as it would allow readers a better understanding of the issues presented and discussed here, starting with the M&M chapter.
E.g:
- presentation of the number of descendants per family (semi-sibling individuals, from the common mother tree), which is unclear in this manuscript (see lines L 151-155), the evolution of the trees and the survival of trees by families [does inbreeding occur?], the height of the trees etc., all these being in fact presented in the work from Baltic Forestry… (e.g. see the Table 1, in that ms.
Title
The formula "Influence of the Genetic Factor and Spatial Structure" in the title is not justified. Moreover, "Genetic Factor" is a notion without obvious arguments in this research. In addition, it is not explained what the authors want to say by this notion or what to be understood using it. The simple comparison of the increases in diameter of the trees from the half-sib families (offspring with a common mother, and unknown father, due open pollination in the stand) is extremely poor for 'genetic factor', 'genetic influence' etc. in a peer-reviewed manuscript. Phenotypic variability of DBH is not ‘genetic factor’. No heritability analysis for the analysed trait - DBH (even if there are models by which one can calculate the heritability in half-sibling families according to mother) and variance decomposition and proportion among trees (in common offspring with the same mother tree) and half-sib families, or molecular analysis (to identify possible DNA differences among seedlings), was performed.
Therefore, the title should be rethought, but also the notion - formula "Genetic Factor" and its sense in the whole manuscript.
Abstract
Lines L 30-34. I recommend simplifying the text of L 30-34: "The target area of our study in 2018 was one of three sites of a unique artificial tree stand included in the EUFORGEN-EUFGIS portal. It was created in 1996 in the Polish part of the Białowieża forest in the Hajnówka district (forestry Wilczy Jar) by Professor A. Korczyk from 10 half-sib families originating from seed collected in the Tisovik tract.". Eliminates: the year (2018, and instead includes the age at which the trees were at the moment of determining DBH); "was one of three sites of a unique artificial tree stand included in the EUFORGEN-EUFGIS portal" (and reformulate it); "created ... by Professor A. Korczyk" (please try to use a concise, accurate, informative but pragmatic style in the whole ms).
L 36: “DBH”. Yes, I totally agree that DBH is "one of the most important silvicultural and breeding traits of trees - diameter", but if you use an abbreviation, you must explain it correctly in a ms. It seems that you did not realize that term 'diameter at breast height' (DBH) does not appear at all in your ms. Remember that a scientific paper (and used notions) must be accessible, explained and understood by non-specialists in the field.
Improve the abstract as proportion and structured form of the manuscript (ms chapters/sections - background, material and methods, results, conclusion). Highlight better what you found as major data, relationships, and your interpretation or main consequences of your findings (conclusions/originality).
Introduction
I suggest you be clearer with some aspects and to better correlate the introduction with the abstract, your subject of study and the results obtained. E.g:
L47: “Since 2012, massive drying of spruce (Picea abies L. Karst.) from insect attack [1,2]…” - If the main pests that have caused the spruce to dry out are bark beetles (Ips typographus L.), please specify here also. Ips typographus L. appears only in the Abstract, but it is desirable to be mentioned also in the full text of the manuscript, in this line if you refer to this pest.
Lines L 55-62: "Taking into account the autochthonous nature of the silver fir from Tisovik, it is reasonable to assume that the A. alba does not pose a threat to biodiversity, but, on the contrary, enhances it, unlike, for example, a very aggressive invasive species, Quercus rubra (L.), which displaces Q. robur (L.) and thus poses a real threat to the stability of forest ecosystems [9,10]. In this case, it is reasonable to assume that the A. alba does not pose a threat to biodiversity, but, on the contrary, enhances it, unlike, for example, a very aggressive invasive species, Quercus rubra (L.), which displaces Q. robur (L.) and thus poses a real threat to the stability of forest ecosystems [9,10].".
Please revise the second half of this paragraph as it is repetitive (you have not revised the paper and the same text is repeated). Please review the entire paper to avoid such errors or other mistakes that should be checked in your self-review.
L74-77:
“It is essential to note that despite its small size - approximately 20 trees at present [7,13,14] – the autochthonous population of silver fir in the Tisovik tract is similar in the degree of polymorphism and heterozygosity compared to the populations from the main distribution area in the Carpathians [15]. This small population shows no inbreeding effect…”
Such information, essential for your study, must be presented much more clearly and at the same time with solid and relevant arguments. It is desirable that the citations / references be represented mainly by the most credible sources, of wide international circulation and easy to find in academic databases (not by regional journals, or in a language with limited circulation in a country; of course, they could be used to strengthen some of the arguments or specific situations).
Except for your reference [15], there are many articles in peer-reviewed journals about the polymorphism and heterozygosity of the populations of Abies alba in the Carpathians.
The issue of inbreeding must be treated much more carefully and connected to your results. The term 'inbreeding' appears in your manuscript only in the form of a citation to a reference (Line 336; reference no. 34, again a title reprinted of a regional journal).
Probably the death and evolution of some trees and families (from the paper published in Baltic Forestry, not shown here), as well as some large differences in trees' growth in diameter could be explained by the very probable inbreeding which follow in the inter-pollinations of some related trees, living on a small area.
L 114-125
The objectives of the study are to be reviewed both for 'the genetic factor' (see the previous comment) and also for the spatial structure factor (i.e., "understand the extent that current spatial structure parameters reflect the age of the stand and the biology of A. alba" it's not really the clearest and most appropriate formulation).
Material and Methods
The M&M chapter is not properly presented and structured, and in addition, it contains ambiguous information in mixed parts, some belonging to the Results chapter.
L129-143 Sub-chapter (sub-section) “2.1. Object of Research”
It is noticed that it was not wanted to repeat some information from ms published in Baltic Forestry (e.g. Table 1, Table 2 - https://www.balticforestry.mi.lt/ojs/index.php/BF/article/view/527), explanation about tree stand, number of seedlings per each family and their evolution by year of observation (1996, 2000, 2004, 2006, 2010, 2018), average diameter of the trunks (diameter at breast height), average height, crown, distance to the nearest tree in the four cardinal points etc. Unfortunately, without some supplemental information, the reader will not clearly understand the context of the current study and some explanation and relevance of the results. A clear presentation of the ten families of half-sib trees, and the number of individuals within each family is absolutely necessary. Probably, the solution would be to find a way to present this information, without plagiarizing the previous manuscript.
Some sentences do not have their rightful place in the M&M section, i.e. L 133-134: "A half-sib family is a collection of plants obtained from seeds from free pollination from one tree." but is justified in the Introduction or Discussion sections.
The sections "2.2. Collection of Field Data and its Standard Processing" and "2.3. Methodological Approach for Determining the Influence of a Genetic Factor" do not have the coherence and clarity necessary to facilitate understanding by readers.
Tables 1, 2, 3 and 4 show results and cannot be placed in the M&M chapter. In addition, their titles are inaccurate or ambiguous (e.g. “Table 1. Statistical indicators of the diameter of the stem by half-sib families”; “Table 2. Diameter classes depending on the genetic factor (??)”; “Table 3. Basic statistical indicators used to group data by class in determining the influence of the factor of spatial structure [12-??? (Do you refer to Table 2 in Baltic Forestry?)]. etc.). Also, their content is not properly correlated with data presentation (or are based on previous work), please see the previous comments.
For a typical quantitative trait such as the trunk diameter of trees, compared to the average, standard deviation appears to be very large (Tables 1, 2 etc.). It is explained that the data were not distributed normally for DBH and Kruskal–Wallis test was used, followed by the Mann–Whitney U test for multiple comparisons.
It will probably be difficult for the reader to understand the large differences between the trees' DBH and phenotypic variability of the trait, without clearer and more obvious arguments (including previous information and the evolution of the number of offspring per family in time, justified discussions of the data and hypotheses, including the probable hypothesis of inbreeding; environmental influence; why some trees have dried up; there are some trees about to dry up and they have been measured; why these large differences between DBH trees etc.).
Results
Please see previous comments (Tables 1, 2, 3 and 4 show results and cannot be placed in the M&M chapter).
The presentation of data in some tables (i.e. "Table 4. Diameter classes depending on the spatial structure") and their repetition in the text (L 206-214) is superfluous.
Other issues:
Unclear title, e.g. “Table 8. Tree diameter by spatial structure sample statistics.”
Unclear data and interpretation, e.g. “Table 5. Pairwise comparisons of сlasses”, “Table 7. Pairwise Comparisons of classes according to spatial structure.”. Why for ‘Comparison groups’ Class 1 / Class 2 and 1 / 3 you used p-value <0.01, but for Class 2 / Class 3 you used as p-value 0.03 (why not <0.05?).
“Таble 9. Correlation between the diameter of trees and indicators of their spatial distribution. Ln is the correlation for the north direction, Ls – south, Le – east, Lw – west, and Lav is the average correlation for all directions.”
Why do you refer to correlation values from a bibliographic resource (“The relationship between the analyzed indicators is weak if the correlation coefficient is less than ± 0.3; from ± 0.3 to ± 0.7 - the relationship is the average strength; if 260 greater than ± 0.7 - the relationship is close [30].”) and not to the number of degrees of freedom in your experience??? The concrete data and the number of analyzed cases decisively influence the significance of the correlations, compared to the indicative values from the specialized literature.
“Table 10. Advantage trees candidates”
Unclear title, understandable for your date presented here. The term 'candidate' (used for 'plus tree') appears only three times in the manuscript (once in the Introduction, here too). Obviously, it is too pretentious to refer to 'candidates', only on the basis of DBH.
“Figure 5. View in 2021 of a fir tree on a half-sib plantation, lagging in the growth of its neighbors in 2015 (marked in green) (photo by A. Marozau).”
The figure and explanation are unclear and the connection between the mentioned years (2021, 2015) and the year in which the study was performed - 2018 (DBH measurements) is difficult to understand.
“Figure 9. Biogroup of old firs in the Polish part of Białowieża Forest District (subdistrict Grudki, subcompartment 498 Сi ) (photo by A. Marozau).”
Is it related to the manuscript and the results from 2018?
Therefore, the manuscript should be carefully reviewed, and the discussions, which are interesting and could be impactful, adjusted accordingly according to the suggestions made above.
Conclusion
None of the conclusions formulated in the first two paragraphs ("The study results suggest that at this particular age, genetics has a more significant influence on the growth of the trees in diameter throughout the whole range of its variation than has the spatial structure"; "Free pollination (father unknown) of a particular mother tree in the Tisovik tract is the basis for the emergence of the half-sibs we studied") is clearly supported by the results and evidence presented. The third conclusion is debatable, especially because the relationship P = G + E + GxE has not been sufficiently well analyzed and deepened genetically, even in terms of interactions (i.e. genotypes, many environmental and ecological elements are missing - general and local factors).
Consequently, the conclusions need to be reformulated and based only on the results obtained, or they must reflect the assumptions of the results.
Reviewer 3 Report
The manuscript makes a two-fold impression. On the one hand, the objectives are quite clear, and the research may be of use for the afforestation projects in the area. On the other, there are two major comments that require the attention of the authors.
First, a constant repetition of the term 'genetic' produces confusion for a reader. The fact is that the authors did not perform a genetic study. The term genetic stems from the word 'gene' and no research on genes are presented in the manuscript. What the authors really studied is the origin effect when it was known from which mother tree the seeds originate. The repetition of the term 'genetic' throughout the text makes a wrong impression. I, therefore, believe that the authors should revise the manuscript taking the use of the term 'genetic' very carefully and probably in more or less in the conjunctive mood.
Second, in lines 173-184, 193-196 the authors describe how they distribute the data into classes. The authors should indicate if the distribution was performed through a purely subjective approach or a formal procedure (like e.g. cluster analysis) was applied. For now, it looks like a fully subjective result. If any other researcher would perform the division into classes in another manner probably he/she would come to some other inferences. So, an indication is required on to what extent the classes appear as an objective result.
Minor comments:
l. 116 - 125: - isn't it necessary to put the particle 'to' into the list?
l. 158: with a tape were measured - were measured with a tape?
Table 6, Table 8: multi - the indication should be defined
l. 250: semi-sibs - same as 'half-sibs'?
l. 261: greater than ± 0.7 - the relationship is close [30]. - probably, the sentence is incomplete
l. 262-263: and with the distances in specific directions positive average strength (Table 9). - the sentence isn't clear, probably, incomplete
l. 344-345: These results suggest that the southern exposure is characterized by the highest intensity of solar radiation - these results do not suggest that; at most, they can support what may be shown with simple instrumental measurements of irradiation.
Figure 6.: The authors should justify why the cross was put in the particular point. It is not the extremum, and I would place it in the rightmost position.
l. 421-422: explain about the same diameter - explain the same diameter?